# Approaches to Nutritional Screening in Patients with Coronavirus Disease 2019 (COVID-19)

**DOI:** 10.3390/ijerph18052772

**Published:** 2021-03-09

**Authors:** Amira Mohammed Ali, Hiroshi Kunugi

**Affiliations:** 1National Center of Neurology and Psychiatry, Department of Mental Disorder Research, National Institute of Neuroscience, Tokyo 187-0031, Japan; hkunugi@ncnp.go.jp; 2Department of Psychiatric Nursing and Mental Health, Faculty of Nursing, Alexandria University, Alexandria 21527, Egypt; 3Department of Psychiatry, Teikyo University School of Medicine, Tokyo 173-8605, Japan

**Keywords:** coronavirus disease 2019/COVID-19, cytokine storm, older adults/elderly, aging/age-related non-communicable diseases, malnutrition/nutritional deficiencies, Nutrition Risk Screening 2002, the controlling nutritional status score/CONUT score, anemia/ferritin, vitamin D, selenium, micronutrients

## Abstract

Malnutrition is common among severe patients with coronavirus disease 2019 (COVID-19), mainly elderly adults and patients with comorbidities. It is also associated with atypical presentation of the disease. Despite the possible contribution of malnutrition to the acquisition and severity of COVID-19, it is not clear which nutritional screening measures may best diagnose malnutrition in these patients at early stages. This is of crucial importance given the urgency and rapid progression of the disease in vulnerable groups. Accordingly, this review examines the available literature for different nutritional screening approaches implemented among COVID-19 patients, with a special focus on elderly adults. After a literature search, we selected and scrutinized 14 studies assessing malnutrition among COVID-19 patients. The Nutrition Risk Screening 2002 (NRS-2002) has demonstrated superior sensitivity to other traditional screening measures. The controlling nutritional status (CONUT) score, which comprises serum albumin level, cholesterol level, and lymphocytes count, as well as a combined CONUT-lactate dehydrogenase-C-reactive protein score expressed a predictive capacity even superior to that of NRS-2002 (0.81% and 0.92% vs. 0.79%) in midlife and elder COVID-19 patients. Therefore, simple measures based on routinely conducted laboratory investigations such as the CONUT score may be timely, cheap, and valuable alternatives for identifying COVID-19 patients with high nutritional risk. Mini Nutritional Assessment (MNA) was the only measure used to detect residual malnutrition and high malnutrition risk in remitting patients—MNA scores correlated with hypoalbuminemia, hypercytokinemia, and weight loss. Older males with severe inflammation, gastrointestinal symptoms, and pre-existing comorbidities (diabetes, obesity, or hypertension) are more prone to malnutrition and subsequently poor COVID-19 prognosis both during the acute phase and during convalescence. Thus, they are in need of frequent nutritional monitoring and support while detecting and treating malnutrition in the general public might be necessary to increase resilience against COVID-19.

## 1. Overview

Coronavirus disease 2019 (COVID-19) is a highly infectious viral disease that results from pulmonary invasion by a beta-coronavirus, known as severe acute respiratory syndrome-coronavirus-2 (SARS-CoV-2) [1]. Even though the disease is asymptomatic in most patients, symptoms of cough, dyspnea, unremitting fever, myalgia, and fatigue commonly occur in specific patient groups; mainly elderly adults and people with chronic disorders such as diabetes mellitus, cardiovascular disorders, obesity, and cancer [2,3,4]. SARS-CoV-2 induces serious adverse effects in these groups including acute respiratory distress syndrome (ARDS), fulminant myocarditis, renal injury, hepatic injury, secondary infections, and mortality [2,5,6,7]. Until now, the global deaths caused by this virus exceeded 2 million [4].

Tissue damage that contributes to multiple organ failure in severe COVID-19 patients is attributed to defective stimulation of the immune system leading to a condition of sustained inflammation—the cytokine storm, which involves excessive and uncontrollable release of inflammatory cytokines [2,3,4]. Symptomatic COVID-19 patients possibly exhibit a baseline state of inflammation, which allows the virus to escape the immune system and take over cellular processes in human cells to promote its replication [8,9,10]. Particularly, it alters the metabolism of key immune cells such as monocytes and macrophages to be highly glycolytic [10]—glucose metabolism dysfunction promotes viral growth [10] and potentiates inflammation [11,12,13]. In addition, the structure and function of major cells responsible for innate immunity get dramatically changed [2,9,14]. For example, lymphocyte and platelet counts decrease considerably in severe conditions while the white blood cell (WBC) count and naïve helper T (Th) cells increase [2,14]. The proliferation of regulatory T cells CD4+ and CD8+ lymphocytes decreases remarkably during the active phase and continues to be low during convalescence [9]. Observed changes in the immune cell composition are an outcome of a defective interaction that entails the conversion of CD4+ cells into Th1 cells. The latter are pathogenic; they secrete high levels of granulocyte-macrophage colony-stimulating factor (GM-CSF) [15]. GM-CSF activates inflammatory cells such as CD14+CD16+ monocytes resulting in excessive production of proinflammatory cytokines [15,16]. Intense alterations in immune cells correlate with increased disease severity and mortality [14,17].

Numerous nutrients are necessary for proper immune functioning [11,12,18,19,20]. Nutritional deficiencies weaken the immune system and increase the invasion, replication, and mutation of viruses [18,21]. Indeed, the pathogenicity of severe acute respiratory syndrome-coronavirus-2 (SARS-CoV-2), the virus that causes coronavirus disease 2019 (COVID-19) [4,13], is associated with an imbalance in several nutritional elements [5,22,23]. Nutritional deficiencies are common among severe and fatal COVID-19, especially elderly adults and patients with age-related comorbidities such as diabetes and cardiovascular disorders [2,5,6,24]. Hypovitaminosis D [5,23], anemia [2,6,25,26,27], iron metabolism dysfunction [2,6], selenium deficiency [22,23], and hypoproteinemia [24,28] are associated with increased levels of proinflammatory cytokines, disease severity, increased admission to the hospital/intensive care unit (ICU), need for mechanical ventilation, and death among COVID-19 patients.

While elderly adults and diseased conditions have a poor baseline nutritional status [11,29,30,31], the cytokine storms associated with COVID-19 as well as disease-related treatments increase the risk for malnutrition [24,32]. Symptoms of vomiting, anorexia, and decreased food intake occur in 25.8% of all symptomatic COVID-19 patients [33], and their prevalence is higher in critical patients [34]. Repurposed antiviral drugs such as hydroxychloroquine and ribavirin cause diarrhea and anemia [35]. The latter also develops as a result of the inflammatory reaction that decreases iron absorption and traps iron in body tissues leading to low levels of circulating iron [32,36]. Hypoproteinemia in COVID-19 patients results from liver injury induced by the cytokine storm [24,37]. Antiviral treatments (e.g., lopinavir/ritonavir) also induce liver injury in a considerable proportion of hospitalized COVID-19 patients [38]. Additionally, bed rest and mechanical ventilation trigger hypoproteinemia by promoting the excessive breakdown of skeletal muscle protein over a short period of time [11,28,39,40].

## 2. Detection of Malnutrition in COVID-19 Patients Is a Challenge

Even though more than 90% of COVID-19 patients who progress to ARDS express at least one nutritional deficiency [23], gold standards for the assessment of nutritional risk (NR) among vulnerable individuals and those who contract COVID-19 are lacking [41,42]. A recent systematic review reports high sensitivity of the Nutritional Risk Screening 2002 (NRS-2002), the Mini Nutritional Assessment (MNA), the MNA-short form (MNA-sf), the Malnutrition Universal Screening Tool (MUST), the Nutritional Risk Index (NRI) for identifying nutritional risk in elderly adults (age range = 65 to 87 years) affected by COVID-19. Among participants, nutritional risk was detected in 27.5% to 100% [42]. The MNA-sf better predicts poor appetite and weight loss; the NRS-2002 better predicts prolonged hospitalization, and the modified Nutrition Risk in the Critically ill (mNUTRIC) score better predicts hospital mortality [34,42,43]. Nonetheless, the authors noted that none of all measures is acknowledged as the best measure for nutritional risk screening in elderly adults with COVID-19 [42]. It is worth mentioning that the indicated review included only four studies from China, sample sizes were small (range = 6 to 182 participants), and a considerable number of relevant studies were not included, which limits the generalizability of the findings [42]. A study comparing the Subjective Global Assessment (SGA), NRS-2002, and MUST with the new Global Leadership Initiative on Malnutrition (GLIM) diagnostic criteria among hospitalized older patients revealed that SGA has the highest sensitivity (96%). Although the sensitivity demonstrated by MUST was less than that of SGA (64%), MUST was the best in terms of specificity and concordance with GLIM criteria (82% and 89%, respectively) [44]. Still, these results may be contextually irrelevant because participants of that study were not COVID-19 patients.

The nutritional status in COVID-19 serves as a prognostic factor—being associated with progression to disease severity and adverse effects (e.g., ICU admission, mechanical ventilation, and mortality). Thus, it is pivotal to properly assess nutritional risk in these patients [41,45]. Because of the intense inflammatory nature of COVID-19, some nutritional biomarkers may not effectively reflect malnutrition [46]. For example, high serum ferritin, which usually portrays increased iron store level, is associated with circulating iron deficiency in COVID-19 [32,46]. Instead, in this context, it is considered a biomarker for the acute phase response that associates with the cytokine storm [2,6]. The situation is further complicated by variations pertaining to gender and different comorbid conditions [41,47,48]. Traditional nutritional screening measures, involving a complete nutritional examination, are only performed by qualified health professionals such as nutritionists or physicians [42]. Given the urgent and rapidly progressing nature of COVID-19 [24,26,41,45], there is less chance to have such sophisticated examinations frequently conducted. Thus, some vulnerable patients may be missed out ending with the development of serious disease adverse effects. Moreover, the traditional nutritional measures could not identify nutritional risk in elderly adults in the ICU [42].

A standardized and proactive nutritional monitoring for elderly adults affected by COVID-19 is necessary to identify nutritionally-frail people who are more prone to poor disease outcomes for a better match with clinical interventions [49]. To bridge the gap, this review explores different approaches to nutritional screening in COVID-19 patients in detail as an attempt to identify the most promising and practical measures, especially those suitable for use among vulnerable individuals such as elderly adults. Studies included in this narrative review were obtained by searching PubMed for investigations assessing malnutrition among COVID-19 patients. The search terms included coronavirus disease-2019, COVID-19, malnutrition, micronutrient deficiency, and nutritional screening. We also hand-searched Google scholar for similar studies. We included studies reporting on the prevalence of malnutrition in COVID-19 patients, either during the acute phase or after recovery, as diagnosed by traditional nutritional measures or nutritional indices. Studies not in English, not reporting a diagnosis of malnutrition, reporting deficiencies of single dietary elements, or reporting only nutritional biomarkers were not included in our synthesis. Out of 197 retrieved studies, 15 studies assessed malnutrition among COVID-19 patients. After excluding a study written in Spanish, 14 studies were summarized in final (for more details, see Appendix A).

## 3. Measures Used for Nutritional Screening in COVID-19 Patients

Table 1 shows that the risk for malnutrition in COVID-19 patients can be assessed by a wide range of measures. Seven studies counted primarily on the traditional screening measures [28,34,48,50,51,52,53]. Some of these studies also included nutritional biomarkers (e.g., total protein and albumin) [28,48,52,53] or anthropometric parameters such as body weight, body mass index (BMI), and calf circumference [34,48,50,51,52]. On the other hand, some studies assessed malnutrition through nutritional indices, which are calculated based on combinations of nutritional biomarkers (e.g., serum albumin and cholesterol), markers of inflammation (e.g., lymphocyte count) [41,47,54], age, comorbidities, etc. [43,48]. Three studies used both traditional measures and calculated indices [34,45,55]. COVID-19 related outcomes addressed in these studies included disease severity [34], COVID-19 complications (such as renal/hepatic injury [45] and muscle dystrophy [41]), admission to the ICU [47], need for oxygen therapy [56], and in-hospital mortality [41,43,45,47,54,55]. One study evaluated the length of hospital stay (LOS), hospital expenses, loss of appetite, and weight loss [34] while many studies evaluated the association between malnutrition and markers of inflammation [28,41,45,53,54,55]. This section elaborates on the findings reported in these studies.

In a study administering MUST, NRS-2002, MNA-sf, and NRI among hospitalized elderly adults with COVID-19, NRS-2002 was the best predictor of malnutrition (85.5%) while MUST had the lowest predictability (41.1%) among the four measures [34]. In another study, NRS-2002 predicted malnutrition in 93% and 100% of severe and critical COVID-19 patients, respectively [28]. Higher NRS-2002 scores were associated with hypoproteinemia and markers of inflammation [28]. GLIM identified malnutrition in 66.4% of ICU-admitted patients. However, the nutritional status was not associated with disease symptoms [53]. mNUTRIC detected high nutritional risk in 61% of COVID-19 patients in ICU, and such a risk was associated with more tissue damage (e.g., ARDS, renal and hepatic injury, etc.) and mortality [43]. NUTRIC is calculated by summing the scores of six parameters: (1) age <50, 50–74, >75 years is scored as 0, 1, 2, respectively; (2) Acute Physiology and Chronic Health Evaluation II (APACHE II) <15, 15–19, 20–27, >28 is scored as 0, 1, 2, 3, respectively; (3) Sequential Organ Failure Assessment (SOFA) <6, 6–9, >10 is scored as 0, 1, 2, respectively; (4) number of comorbidities of 0–1 and >2 is scored as 0, 1, respectively; (5) LOS before ICU admission of 0 and ≥1 days is scored as 0, 1, respectively; (6) interleukin (IL)-6 of 0–399 and ≥400 µ/mL is scored as 0, 1, respectively. Scores of APACHE II and SOFA are considered on ICU admission. mNUTRIC does not include IL-6 in its estimation, and a score >5 indicates a high NR [43].

Several studies portray consistent changes in nutrition-related biomarkers in severely and critically ill COVID-19 patients [28,48,53]. Severely ill patients experience dyspnea, respiratory rate ≥30/min, blood oxygen saturation ≤93%, the partial pressure of arterial oxygen to fraction of inspired oxygen ratio <300 mmHg, and lung infiltrates >50% within 24 to 48 h. Critically ill patients may need ICU admission and mechanical ventilation secondary to respiratory failure, shock, disseminated coagulopathy, and multiple-organ failure [57]. Compared with severely ill patients, critically ill patients express significantly lower levels of total protein, serum albumin, and prealbumin as well as significantly higher levels of serum urea nitrogen, creatinine, glucose, and total bilirubin [28,37]. Nutritional markers in severe/critical COVID-19 patients correlated with NRS-2002 [28] and NRI [56]. Likewise, among remitting patients, MNA scores correlated with nutritional markers [52]. In a study involving 2623 COVID-19 patients (median age = 64 years), non-critical, critical, and fatal patients, in order, expressed significant hypoalbuminemia on admission (38.2%, 71.2%, and 82.4%), which increased during hospitalization (45.9%, 77.7%, and 95.6%) [24]. Lower serum albumin levels (<10 g/L) demonstrated a higher risk for ICU admission (odds ratio (OR = 0.31, 95%CI: 0.1–0.7, *p* < 0.01) regardless of age and C-reactive protein (CRP) levels [48,53]. Additionally, albumin levels lower than 29.6 g/L could independently predict mortality in COVID-19 patients admitted to the ICU [58,59].

Weight loss >10% of initial body weight is associated with malnutrition in recovering COVID-19 patients both with a history of hospitalization [34,50,51,52] and home-treatment (9.6% vs. 5.3%, *p* = 0.41) [51]. Skeletal muscle is a key structure involved in the regulation of glucose metabolism and overall homeostasis [60]. Skeletal muscle injury signaled by myalgia and elevated markers of muscular dystrophy such as myoglobin, creatine kinase (CK), and lactate dehydrogenase (LDH) is common in COVID-19 patients [61,62]. Muscle and weight losses are extensive in COVID-19 patients during their stay at ICU [39] and following ICU discharge [52]. Weight loss during ICU stay is likely to be related to muscle breakdown rather than fat loss [39]. Obese and overweight patients represent a majority of ICU-admitted COVID-19 patients [50]. In addition to muscle deconditioning due to immobility associated with the prolonged prone position in ICU [63], obesity stimulates muscle fiber shrinkage through an inflammatory mechanism [60]. In remitting patients, weight loss was associated with high CRP, longer LOS, and disease duration independent of age, sex, pre-existing comorbidities, and most of the biochemical parameters upon admission [51]. Among discharged recovering patients, 69%, 38%, 30.1%, and 9.5% experienced fatigue, abnormal chest radiographs, and persistently elevated d-dimer and CRP, respectively while 9% of patients were deteriorating [64].

The cytokine storm evokes insulin resistance resulting in hyperglycemia—a key feature in COVID-19 because SARS-CoV-2 directs host metabolism for the sake of its replication [65]. Hyperglycemia furthers inflammation and oxidative stress through the activation of the receptor of activated glycation end products [11,31]. Oxidative stress, cytokines, and insulin resistance activate catabolic signaling resulting in muscle protein degradation and muscle loss [12,60,62], which is evident in COVID-19 patients regardless of their status of hospitalization or ICU admission [62]. However, BMI alone is not considered a suitable indicator of malnutrition in COVID-19 [42]. A combination of nutritional biomarkers and anthropometric measures in certain clinical conditions can be used to signal the risk of malnutrition in COVID-19 patients [48,66]. A score based on a combination of a diagnosis of diabetes, low calf circumference, and low serum albumin level is reported as an independent risk factor for malnutrition in COVID-19 patients [48]. This combined score correlates with high MNA scores [48].

Several indices have been developed as easy-to-conduct measures of malnutrition based on combining certain laboratory parameters together or with anthropometric measures. The Prognostic Nutritional Index (PNI) is calculated as 10 × serum albumin (g/dL) + 0.005 × total lymphocyte count (mm^3^). PNI > 38, PNI of 35–38, PNI < 35, in order, reflect normal nutrition, moderate, and severe risk for malnutrition [54,55]. The controlling nutritional status (CONUT) score—calculated from lymphocyte count, total cholesterol, and serum albumin—is used for nutritional screening [41,45,54]. Lymphocyte counts: ≥1.600, 1.200–1.599, 0.800–1.199, <0.800 × 109/L are scored as 0, 1, 2, 3, respectively. Serum albumin: ≥3.5, 3.0–3.49, 2.5–2.99, <2.5 g/dL are scored as 0, 2, 4, 6, respectively. Serum cholesterol: ≥180, 140–179, 100–139, <100 mg/dL are scored as 0, 1, 2, 3, respectively. Summing every separate score of the aforementioned parts results in the CONUT score [45]. CONUT scores of 0–1, 2–4, 5–8, >9 denote normal nutrition, mild, moderate, and severe malnutrition, respectively [54]. The Nutritional Risk Index (NRI) is calculated based on serum albumin and recent bodyweight loss:  NRI =  (1.519  ×  serum albumin, g/L)  +  0.417  ×  (present weight/usual weight  ×  100) [34].

In COVID-19 patients (around 50% > 65 years), the CONUT score classified 46.3% and 39.9% of patients as having mild and moderate-severe malnutrition, respectively [41]. Compared with normal and mildly malnourished patients, moderate-severe malnourished patients were likely to be older, diabetic, or hypertensive, and they significantly had higher white blood cell and neutrophil counts, higher CRP, LDH, aspartate aminotransferase (AST), CK myocardial isoenzyme, and lower levels of albumin, total cholesterol as well as lymphocyte and platelet counts [41,45]. Patients with a higher CONUT score had higher development of acute cardiac injury (44.6%) [41], lower survival to discharge, and higher all-cause mortality (adjust OR = 1.4 95% CI 1.089–1.825). Cardiac injury, dyspnea, high CRP, LDH, and advanced age were associated with mortality [41,45]. Figure 1 presents a schematic comparison of different nutritional screening measures used in COVID-19 patients.

The CONUT score predicted the prognosis of COVID-19 patients with a sensitivity and specificity of 74.1% and 72.0%, respectively [41]. The CONUT score predicted mortality better than PNI: compared with non-survivors, COVID-19 survivors had higher PNI (43.95 vs. 36.95, *p* < 0.001) but lower CONUT score (3 vs. 6, *p* < 0.001) [45]. Interestingly, comparing the CONUT score with NRS-2002 revealed that the CONUT score demonstrates a slightly higher prognostic potential for COVID-19 than NRS-2002 as indicated by the area under the ROC curve (AUC) of 0.813 and 0.795, respectively [45]. A combined model of CONUT, LDH, and CRP had even a higher prognostic capacity (AUC = 0.923, Z = 3.5210, *p* < 0.001) [45]. The predictive capacity of malnutrition in elderly adults with COVID-19 (mean age = 71.7  ±  5.9 years) by NRI was lower than that of NRS-2002 (71.6% vs. 85.8%) [34]. NRS-2002 was reported to have superior sensitivity to other equivalent measures such as MUST and MNA-sf [34,42]. In the meantime, the predictive capacity of NRS-2002 is less than that of the CONUT score and the combined CONUT-LDH-CRP (0.795%, vs. 0.813 and 0.923, respectively) in midlife and old COVID-19 patients (mean age = 58 years, range = 41–70 years) [45]. Moreover, higher CONUT scores correlate with age [41,45,55]—COVID-19 patients ≥61 years old with high CONUT score exhibit a 6.2-fold higher risk for adverse outcomes compared with counterparts with low CONUT score (relative risk = 6.191, 95% CI: 1.431–26.785) [47]. According to Table 1, the majority of COVID-19 patients in most studies were elderly adults. Comorbidities were also reported in several studies. Therefore, the CONUT score and the combined CONUT-LDH-CRP may be better used for predicting malnutrition and COVID-19 prognosis in elderly adults given that less cost and skills are needed to obtain these measures. It is worth mentioning that the cutoff points of the CONUT score were inconsistently defined in some studies. In one study, a group of moderate-severe cases was defined based on a CONUT score > 5 [41], while another study identified high malnutrition by a CONUT score of 5–12 [47]. In a third study, severe malnutrition was based on a CONUT score > 9 [54].

## 4. Critical Risk Factors for Malnutrition in COVID-19

Current guidelines emphasize the importance of referring COVID-19 with high nutritional risk to a dietitian to facilitate adequate nutritional support [67]. Therefore, it is necessary to identify patients with the highest risk for malnutrition. The elders are generally prone to malnutrition due to age-related physiological changes in the gastrointestinal (GI) tract (e.g., poor dentation, swallowing, taste/smelling, delayed gastric emptying, and gut-microbiome deficiency) in addition to several psychological, cognitive, social, and financial limitations that hinder their intake of adequate dietary elements [68,69,70,71]. Old age is associated with subclinical chronic inflammation and increased incidence of age-related illnesses such as cardiovascular diseases [20,31]. In 12 out of 14 studies, the average/median age of participants was 55 years or above (Table 1) denoting that old age is a critical risk factor for malnutrition in COVID-19. In six studies, males accounted for more than 60% of the participants (Table 1). Male gender is associated with increased occurrence of nutritional deficiency and adverse effects of COVID-19 [41,47]. According to Table 1, most nutritional biomarkers/measures correlated with biomarkers of inflammation [28,41,55,56], weight loss [34,51,52,56], disease severity, prolonged LOS [28,34], preexisting comorbidities (e.g., diabetes [41,47,48,50], obesity [50,52,56], and hypertension [41,47,48]), acute myocardial injury [41,43], ICU admission [53], poor appetite [34], old age [47], incidence of ARDS, secondary infection, shock, and use of vasopressors [43]. Antibiotics, antiviral drugs, and bowel invasion by SARS-CoV-2 induce GI symptoms such as diarrhea, vomiting, and anorexia [33,34,35,54,61]. Different GI symptoms are reported in 61.2% of hospitalized COVID-19 patients [72]. These symptoms may be exacerbated by the cytokine storm leading to severe reductions in food intake and increased loss of nutrients. GI symptoms develop in COVID-19 patients before hospital admission [54], and they persist after recovery in a considerable proportion of patients [61]. Accordingly, old age, severe inflammation, GI symptoms, weight loss, pre-existing comorbidities (especially obesity and diabetes), and developing organ failure, particularly in males may be the most critical risk factors for malnutrition signifying a need for careful nutritional monitoring and support.

## 5. Identifying Malnutrition in the General Public during COVID-19 Outbreak Is a Necessity

Malnutrition is an inflammatory condition that occurs simultaneously in chronic diseases such as obesity, diabetes, and cancer as well as in apparently well individuals who adopt unhealthy lifestyles such as lower physical activity and unhealthy (e.g., western) diet [23]. Retrospective data denote a lower incidence of COVID-19 among individuals receiving dietary supplements [73]. Therefore, detecting and treating malnutrition in community-dwelling non-COVID-19 patients and the general public may be essential to increase immune resilience against SARS-CoV-2 [42,74]. Nutritional screening for vulnerable individuals in the community (e.g., the elderly and diseased conditions) may be conducted through digital channels to lower the occurrence of infection due to unnecessary exposure [63,67,75]. Simple remote nutritional screening tools such as Remote-Malnutrition APP (R-MAPP) can be conducted through phone calls. The R-MAPP comprises two measures: MUST, which assesses nutritional risk, and a 5-item questionnaire—Strength, Assistance with walking, Rise from a chair, Climb stairs and Falls (SARC-F), which assesses physical frailty/sarcopenia (skeletal muscle loss and poor physical performance) [75].

Evaluating the condition of skeletal muscle represents an integral part of nutritional screening during the current COVID-19 crisis [51,60]. Reduced levels of physical activity during the COVID-19 outbreak are globally reported, and they correlated with increased prevalence of frailty and falls, especially among the elderly [76,77,78,79]. Physically frail individuals are highly prone to COVID-19; they also progress to severe states when they contract the disease [60,80,81]. COVID-19 also triggers muscle loss as noted above, and the assessment of muscle condition may facilitate the identification of vulnerable individuals [60,62,82]. Readers interested in different methods of assessing skeletal muscle loss, including that occurring in COVID-19 patients, are referred to this comprehensive resource [82].

## 6. Current Knowledge on the Management of Malnutrition in COVID-19

Nutritional guidelines for COVID-19 patients emphasize the use of food-based strategies, oral nutritional supplements, referral to a dietitian, and use of efficient algorithms to provide nutrition for the first 5–7 days in lower-nutritional-risk patients and individualized care for high-nutritional-risk patients [63,67]. Clinical trials show that ICU-admitted patients with sepsis or low BMI who also express high nutritional risk demonstrate significant reductions in 28- and 60-day mortality following an early increase of energy (≥25 kcal/kg) and protein intake (≥1.2 g/kg) [83] or 10% greater protein and energy adequacy [84]. The opposite was not true in patients with low nutritional risk [83]. The proper delivery of nutrients to critical COVID-19 patients may be achieved via prompt initiation of volume-controlled, higher-protein enteral formula and monitoring gastric residual volume [63]. Adherence to fasting guidelines is necessary to ensure adequate enteral nutrition delivery and to reduce the duration of feed breaks [85]. Indirect calorimetry should be avoided because it puts healthcare professionals at risk for infection secondary to aerosol exposure [63].

Rapid communication of malnutrition risk at discharge between settings is necessary to ensure continuity of nutritional care, which should be an integral aspect of rehabilitation pathways for patients recovering from COVID-19 [67]. Indeed, MNA detected malnutrition and high risk of malnutrition in 14.6% and 65.9% of patients following discharge from ICU [52], in 6.6% and 54.7% of remitting patients attending follow up care [51], and in 5.4% and 57.3% of recovering patients 100 days after discharge from the hospital [50]. Some elements in bioactive foods (e.g., bee products) foster the immune system and interact with SARS-CoV-2 and its related host receptor, resulting in lower viral tropism and decreased viral load [4,86]. Bee products, are reported to counteract anabolic resistance in old age, suppress catabolic genes, and foster anabolism in malnourished old rodents [11]. Therefore, supplementing malnourished patients with these foods may facilitate recovery [4,86].

## 7. Conclusions

Among symptomatic COVID-19 patients, older males with high levels of inflammatory mediators, GI symptoms, a diagnosis of diabetes or hypertension, multiple organ failure are more prone to malnutrition. A poor nutritional status predicts a COVID-19 prognosis. Among traditional nutritional screening tools, NRS-2002 seems to be the most suitable measure for detecting malnutrition in COVID-19 patients. Serum albumin level alone may sufficiently reflect malnutrition in hospitalized COVID-19 patients, especially in highly vulnerable individuals. Because of the rapidly progressing nature of COVID-19, the CONUT score and a combined CONUT-LDH-CRP score may represent reliable and convenient alternatives for traditional nutritional screening in COVID-19 patients, especially in elderly adults. Assessing nutritional risk in recovering COVID-19 patients and in the general public via remote channels facilitates chances for treating nutritional deficiencies, which may have implications for improving immune functioning and reducing the risk for COVID-19 infection.

## Figures and Tables

**Figure 1 ijerph-18-02772-f001:**
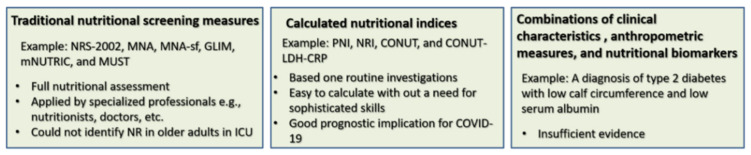
Schematic summary comparing different measures used for nutritional assessment in COVID-19 patients. NRS-2002: Nutrition Risk Screening 2002, GLIM: Global Leadership Initiative on Malnutrition, mNUTRIC: modified Nutrition Risk in the Critically ill, MNA-sf: Mini Nutrition Assessment Shortcut, MUST: Malnutrition Universal Screening Tool, NRI: Nutrition Risk Index, PNI: Prognostic Nutritional Index, CONUT: Controlling Nutritional Status, LDH: lactate dehydrogenase, CRP: C-reactive protein, ICU: intensive care unit, NR: nutrition risk.

**Table 1 ijerph-18-02772-t001:** Nutritional assessment measures in COVID-19 and their correlation with disease severity and prognosis.

Sample Size	Age (Years)	Male Gender	Nutritional Measure	COVID-19 Outcomes	Malnutrition Prevalence	Malnutrition Association with COVID-19 Outcomes	Ref.
141	71.7 ± 5.9	48.2%	NRS-2002, MUST, MNA-sf, NRI	LOS, hospital expenses, appetite, disease severity, weight change	Malnutrition was identified by NRS-2002, MUST, MNA-sf, NRI in 85.8%, 41.1%, 77.3%, and 71.6% of patients, respectively.	Patients high on NRS 2002, MNA-sf, and NRI had significantly longer LOS, higher hospital expenses, poor appetite, disease severity, and more weight loss.	[34]
136	Median age = 69 (IQR: 57–77)	63%	mNUTRIC	Mortality within 28 days of ICU admission	Malnutrition was identified in 61% of critically ill patients.	Compared with low NR patients, malnourished patients had higher mortality (87% vs. 49%, *p* < 0.001), the higher probability of death at ICU 28-day (adjusted HR = 2.01, 95% CI: 1.22–3.32, *p* = 0.006), higher incidence of ARDS, acute myocardial injury, secondary infection, shock, and use of vasopressors.	[43]
114	59.9 ± 15.9	60.5%	GLIM	Clinical, radiological, and biological characteristics of COVID-19 patients	Moderate and severe malnutrition developed in 23.7%, and 18.4% in the whole sample, and in 66.7% of patients in the ICU.	GLIM correlated with lower albumin level and increased ICU admission regardless of age and CRP level.	[53]
413	60.3 ± 12.7	51%	NRS-2002	BMI, inflammatory and nutritional markers	Among all patients, severe, and critical patients, moderate malnutrition developed in 76%, 84%, and 38% of patients, respectively while severe malnutrition developed in 16%, 7%, and 62% of patients, respectively.	High NRS-2002 scores in critically ill patients correlated with inflammatory and nutrition-related markers, LOS, and a higher risk of mortality.	[28]
182	68.5 ± 8.8	36%	MNA	Comorbidities, BMI, calf circumference, albumin, hemoglobin, and lymphocyte counts	Malnutrition and risk of malnutrition in developed in 52.7% and 27.5% of patients, respectively.	A score comprising a combination of diabetes mellitus, low calf circumference, and low albumin is an independent risk factor for malnutrition.	[48]
348	66 (range = 56 to 73)	52%	CONUT	Inflammation and malnutrition markers, mortality, muscle dystrophy	Mild and moderate-severe NR were identified in 46.3% and 39.9% of patients, respectively	Moderate-severe malnutrition correlated with age, inflammation and nutrition markers, the development of acute cardiac injury, and all-cause mortality.	[41]
429	48.3% > 61	65.7%	CONUT	Clinical condition and COVID-19 adverse effects (ICU admission and all-cause death).	Malnutrition was identified in 65.7% of patients.	High CONUT score correlated with old age, diabetes, and hospital admission. Older adults with a high CONUT score had a 6.2 times higher risk of adverse outcomes. Gender, age, hypertension, and urinary erythrocytes were the key factors affecting adverse outcomes. High sensitivity and specificity of the CONUT on the ROC curve.	[47]
295	58 (44–69)	52.5%	GNRI, PNI, CONUT	Nutritional, inflammatory, and renal biomarkers, clinical data, and in-hospital death	Moderate and severe NR in critically ill patients were 10% and 30% on the PNI score and 34.6% and 30.8% on the CONUT score	Critically ill patients had significantly lower albumin levels and higher blood urea nitrogen and serum creatinine, CRP, IL6 than severe or mild/moderate patients (*p* < 0.0001). Baseline nutritional status correlated with in-hospital mortality. Good prognostic implication of GNRI and CONUT score on the ROC curve	[55]
245	Median age = 55	46.5%	PNI and CONUT	In-hospital mortality, clinical data, laboratory, and nutritional biomarkers.	Moderate and severe NR were identified in 12.7% and 12.2% on the PNI score and in 23.7% and 2.8% of patients on the CONUT score.	CONUT score (OR = 3.371,95% CI 1.124–10.106, *p* = 0.030) and PNI (OR = 0.721, 95% CI 0.581–0.896, *p* = 0.003) were independent predictors of all-cause death at an early stage. Higher PNI was an independent risk predictor of in-hospital death (OR = 24.225, 95% CI 2.147–273.327, *p* = 0.010).	[54]
442	58 (41–70)	46.6%	CONUT and NRS-2002	In-hospital mortality, markers of inflammation, nutrition, renal, and liver function, COVID-19 complications	CONUT identified severe malnutrition in 7.6% of non-survivors.	In adjusted analysis, CONUT (*p* = 0.002), LDH (*p* < 0.001), CRP (*p* = 0.020) were risk factors of mortality in COVID-19 patients. Better prognostic potential of CONUT and combined CONUT-LDH-CRP than NRS-2002.	[45]
108	62 ± 16	62.9%	NRI, BMI, 5% or 10% weight loss in the previous month or 6 months	Need for nasal oxygen, markers of inflammation, and nutrition.	NRI identified malnutrition and risk for malnutrition in 38.9% and 84.9% of patients.	NRI scores correlated with inflammation; lower plasma levels of proteins, albumin, prealbumin, and zinc, and the need for oxygen therapy.	[56]
41	55 (19–85)	51.2%	MNA	BMI, weight loss, anemia, and serum levels of Ca, Zn, Mg, albumin, and vitamin D.	MNA identified malnutrition and risk for malnutrition in 14.6% and 65.9% of ICU-discharged patients.	Weight loss in 61% (>10% of body weight in 26.2%) of patients. Hypoalbuminemia, hypoproteinemia, hypocalcemia, anemia, hypomagnesemia, and hypovitaminosis D were detected in 19.5%, 17.1%,19.5%, 34.1%, 12.2%, and 51.2% of patients, respectively.	[52]
185	57 (48–67)	65.5%	MNA	Need for follow-up due to dyspnea, tachypnea, new-onset cognitive impairment, and post-traumatic stress.	MNA identified malnutrition and risk for malnutrition in 5.4% and 57.3% of patients, 100 days following discharge from the hospital or ICU.	BMI and ≥33 Kg/m^2^, arterial oxygen partial pressure to fractional inspired oxygen ratio < 324, age > 63 years, diabetes, and non-invasive ventilation highly predicted the need for follow-up.	[50]
213	Median age = 59 (49.5–67.9)	66%	MNA	Appetite, weight loss, and inflammation biomarkers.	MNA identified malnutrition and risk for malnutrition in 6.6% and 54.7% of remitting patients, following discharge from the hospital or treatment at home.	High risk of malnutrition among hospital and ICU admitted patients. Weight loss > 10% of initial body weight in hospitalized and home-treated patients (9.6% vs. 5.3%, *p* = 0.41) was associated with high CRP, renal injury, longer LOS, and disease duration independent of age, sex, pre-existing comorbidities, and most of the biochemical parameters upon admission.	[51]

Abbreviations: NRS-2002: Nutrition Risk Screening 2002, MUST: Malnutrition Universal Screening Tool, MNA-sf: Mini Nutrition Assessment Shortcut, NRI: Nutrition Risk Index, GLIM: Global Leadership Initiative on Malnutrition, GNRI: Geriatric Nutritional Risk Index, PNI: Prognostic Nutritional Index, CONUT: Controlling Nutritional Status, mNUTRIC: modified Nutrition Risk in the Critically ill, ROC: receiver operating characteristic curves, ICU: intensive care unit, LOS: length of stay, BMI: body mass index, ARDS: acute respiratory distress syndrome, LDH: lactate dehydrogenase, CRP: C-reactive protein, NR: nutritional risk.

## Data Availability

Not applicable.

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
