# Peer review of "Approaches to Nutritional Screening in Patients with Coronavirus Disease 2019 (COVID-19)"

_ijerph, 2021, doi:10.3390/ijerph18052772_

Round 1

Reviewer 1 Report

Ali and Kunugi present a COVID-19-based, mini-review discussing recent standardized screening protocols to identify nutritional nutritionally frail people who are susceptible to other diseases and may be appropriate candidates for nutritional-based clinical interventions.  The authors evaluated traditional nutritional screening measures, including the NRS-2000, MNA, MNA-sf, GLIM, and MUST and calculated nutritional indices such as mNUTRIC, PNI, NRI, CONUT, and CONUT-LDH-CRP. This evaluation included also clinical endpoint measurements such as circulating albumin and total lymphocyte counts. In addition, mortality, possibly affected by COVID-19, was addressed. The authors concluded that the NRS-2000, serum albumin alone, and the CONUT index are effective parameters for screening nutritional deficiencies in COVID-19 patients.

This mini-review was well presented and relevant. Some minor comments:

  • Please state how many articles that were originally found with the query terms and what criteria you used to select the final 10.
  • Please replace “old” adults with “elderly.”
  • Please be sure to use only one decimal place after the decimal. You do not need more than one, especially in your reporting of mean ages.
  • Please be sure to be consistent with “CONUT.” You used “COUNT” rather than “CONUT” in many locations throughout.

Author Response

Manuscript ID: ijerph-1082973.

Title: Approaches to Nutritional Screening in Patients with Coronavirus Disease 2019 (COVID-19).

Response to Comments of Reviewer

First of all, we would like to thank the reviewer for the precious time, hard effort, and sincere advice that he/she had kindly given to make this manuscript better. We have addressed the comments line-by-line as shown below. Replies come underneath in red.

  • Please state how many articles that were originally found with the query terms and what criteria you used to select the final 10.

Authors’ response: Honestly, it was not originally a systematic search—we retained relevant studies from several search hits without keeping track of the search outcomes. Based on the comments of reviewer 3, we had to search for more studies. So, we updated our search keeping it systematic this time. We have described the search strategy and criteria of selection of studies in detail as reviewer 1 indicated (lines 141-150).

  • Please replace “old” adults with “elderly.”

Authors’ response: Yes, we have replaced all the instances of “old” adults with “elderly.”

  • Please be sure to use only one decimal place after the decimal. You do not need more than one, especially in your reporting of mean ages.

Authors’ response: Yes, we used only one decimal point.

  • Please be sure to be consistent with “CONUT.” You used “COUNT” rather than “CONUT” in many locations throughout.

Authors’ response: Thank you very much, we have corrected all instances in the text and the figure.

We hope that we have satisfactorily modified the manuscript and that the revised version will be suitable for publication.

Best regards,

Reviewer 2 Report

Severe patients withCOVID-19 need for frequent nutritional monitoring. Therfore, simple measures are needed.  It is very important and urgent to consider which nutrition screening measures are most suitable for early diagnosis of malnutrition in these patients.

The authors found that simple measures based on routinely conducted laboratory investigations such as the CONUT score may be timely, cheap, and valuable alternatives for identifying COVID-19 patients with high nutritional risk.

I think it is a very important and useful result.

Author Response

Manuscript ID: ijerph-1082973.

Title: Approaches to Nutritional Screening in Patients with Coronavirus Disease 2019 (COVID-19).

Response to Comments of Reviewer

First of all, we would like to thank the reviewer for the precious time, hard effort, and supportive comment. We have expanded the manuscript based on an updated search.

We hope that we have satisfactorily modified the manuscript and that the revised version will be suitable for publication.

Best regards,

Reviewer 3 Report

The article titled "Approaches to Nutritional Screening in Patients with Corona- 2 virus Disease 2019 (COVID-19)" authored by Amira Mohammed Ali and Hiroshi Kunugi provides a review of nutritional screening approaches to COVID-19 patients with a particular focus on older subjects. Although this work resumes the data presents in literature, the manuscript appears incomplete e with some lacks in the data discussion.

  1. Nutritional screenings such as NRS-2002 could be difficult to applicate from not specialized professionals, and their incorrect application dramatically reduces the clinical usefulness of these tests. According to your experience and the literature available so far, what are the most important indices to identify patients with the highest risk? Identifying the most critical risk factors can help clinicians identify patients most at risk by referring them to a nutritionist or dietician's specialist evaluation.
  2. Evaluation of malnutrition in COVID-19 patients is undoubtedly essential in estimating the prognosis of the individual patient. Would the malnourished patient need a different and more personalized clinical approach to improve the prognosis? I would suggest discussing these possible clinical indications for a correct application of nutritional screenings that goes well beyond the simple identification of a worse prognosis.
  3. Are there any data describing malnutrition incidence at discharge after hospitalization for COVID-19 using these screening tests? Should the management of survivors' convalescence also consider nutritional status? 
  4. In addition to the symptomatic patient, could nutritional screenings also help assess the risk in larger sections of the population? Is there data to support the use of these tests even in a healthy patient? I would suggest an evaluation of that effect in order to describe a broader public health approach. 

Author Response

Manuscript ID: ijerph-1082973.

Title: Approaches to Nutritional Screening in Patients with Coronavirus Disease 2019 (COVID-19).

Response to Comments of Reviewer

First of all, we would like to thank the reviewer for the precious time, hard effort, and insightful advice that he/she had kindly given to make this manuscript better. We have addressed the comments line-by-line as shown below. Replies come underneath in red.

  1. Nutritional screenings such as NRS-2002 could be difficult to applicate from not specialized professionals, and their incorrect application dramatically reduces the clinical usefulness of these tests. According to your experience and the literature available so far, what are the most important indices to identify patients with the highest risk? Identifying the most critical risk factors can help clinicians identify patients most at risk by referring them to a nutritionist or dietician's specialist evaluation.

Authors’ response: Based on the available literature, we attempted to identify the most critical risk factors, which can help clinicians identify patients most at risk for referring them to a nutritionist or dietician's specialist evaluation (Lines 300-328).

  1. Evaluation of malnutrition in COVID-19 patients is undoubtedly essential in estimating the prognosis of the individual patient. Would the malnourished patient need a different and more personalized clinical approach to improve the prognosis? I would suggest discussing these possible clinical indications for a correct application of nutritional screenings that goes well beyond the simple identification of a worse prognosis.

Available guidelines and recommendations

Authors’ response: Based on this comment, we have summarized data from nutritional guidelines for COVID-19 patients linking them to findings from clinical trials reporting on nutritional support for malnourished patients (including those with sepsis) in ICU (Lines 355-379).

  1. Are there any data describing malnutrition incidence at discharge after hospitalization for COVID-19 using these screening tests? Should the management of survivors' convalescence also consider nutritional status? 

Authors’ response: Thank you very much for raising such an important issue. Based on this comment, we updated our search and retrieved few studies including nutritional assessment in recovering COVID-19 patients following discharge from the ICU/hospital or a long time (100 days) after remission. The findings highlight the importance of managing nutritional deficiency as an aspect of survivors' convalescence care. We have included these studies in Table 1 and added a corresponding summary paragraph in the text (Lines 372-375).

  1. In addition to the symptomatic patient, could nutritional screenings also help assess the risk in larger sections of the population? Is there data to support the use of these tests even in a healthy patient? I would suggest an evaluation of that effect in order to describe a broader public health approach. 

Authors’ response: Baseline malnutrition entails high immune vulnerability to COVID-19; therefore, proper identification and management of nutritionally deficient individuals may boost the immune system and lower the risk for COVID-19. The intake of dietary supplements is associated with decreased risk for COVID-19. Remote nutritional screening has been proposed by researchers to identify nutritionally vulnerable individuals (including “healthy patients”) in primary care. We have discussed this issue in the revised version (Lines 329-354).

We hope that we have satisfactorily modified the manuscript and that the revised version will be suitable for publication.

Best regards,

Round 2

Reviewer 3 Report

The authors significantly improved the manuscript. The new version of the paper deserves to be published. Excellent job.

This manuscript is a resubmission of an earlier submission. The following is a list of the peer review reports and author responses from that submission.